# Adaptive Interaction Control of Compliant Robots Using Impedance Learning

**DOI:** 10.3390/s22249740

**Published:** 2022-12-12

**Authors:** Tairen Sun, Jiantao Yang

**Affiliations:** School of Health Science and Engineering, University of Shanghai for Science and Technology, Shanghai 200093, China

**Keywords:** impedance learning, adaptive control, compliant robot, impedance control

## Abstract

This paper presents an impedance learning-based adaptive control strategy for series elastic actuator (SEA)-driven compliant robots without the measurement of the robot–environment interaction force. The adaptive controller is designed based on the command filter-based adaptive backstepping approach, where a command filter is used to decrease computational complexity and avoid the requirement of high derivatives of the robot position. In the controller, environmental impedance profiles and robotic parameter uncertainties are estimated using adaptive learning laws. Through a Lyapunov-based theoretical analysis, the tracking error and estimation errors are proven to be semiglobally uniformly ultimately bounded. The control effectiveness is illustrated through simulations on a compliant robot arm.

## 1. Introduction

Safety in robot–environment interaction is of significant value and can be improved by passive compliant devices. As a popular compliant device, a series elastic actuator (SEA) is developed by introducing elastic elements between the motor and the load and can bring some benefits including low output impedance, tolerance to shocks, and energy efficiency [1,2,3]. The introduction of SEAs in robots improves interaction compliance to some extent but (1) it cannot root out the conflict between high robot stiffness and the requirement of high compliance, and (2) the compliant actuators have bad adaptability and limited applications since SEAs make robots behave only in a certain impedance. The compliance of SEA-driven robots should be further improved by the regulation of robot impedance using active compliance control.

As one of the most popular compliance control approaches, impedance control proposed by Hogan in the 1980s [4] provides interaction compliance through a dynamical relationship between the position and interaction force. To date, extensive impedance control strategies for rigid-link robots have been developed based on adaptive learning [5,6,7], sliding mode [8], neural networks [9,10,11], and so on. For impedance control implementation, one significant problem to be solved is the determination of the desired robot impedance, which is highly dependent on environmental impedance. Although a variety of methods, including least-squares techniques and programming by demonstration [12,13,14], were developed for impedance learning, the impedance controllers based on these impedance learning methods were usually designed without stability guarantees. Recently, model-based impedance learning control strategies [15,16,17] were developed for robot–environment interactions and validated in repetitive tasks with stability guarantees. The control approach can provide variable impedance regulations for robots without the requirement of interaction force sensing. However, the existing model-based impedance learning controllers mainly focus on rigid-link robots. The extension of model-based impedance learning control to SEA-driven compliant robots is not direct since the introduction of an SEA significantly increases control design complexity and turns the control system into a fourth-order underactuated system from a second-order fully actuated system.

Based on the above analysis, designing model-based impedance learning control for SEA-driven robots can exploit the advantages of passive compliant devices and active compliance control to improve robot–environment interaction performance, but to date, no results on this topic have been produced.

In this paper, stability-guaranteed adaptive control using model-based impedance learning is proposed for SEA-driven robots with fourth-order underactuated systems. Impedance parameters of interaction forces and model uncertainty parameters are estimated using differential adaptation laws updated by tracking errors. In the control design, the command filter-based adaptive backstepping approach is used to decrease computational complexity and avoid the requirement of the high derivatives of the robot position in the backstepping control of SEA-driven robots. We prove the semiglobal stability of the closed-loop control system theoretically and illustrate the control effectiveness through simulations on a SEA-driven robot arm. The proposed control strategy can be applied to categories of robot–environment interactions including robot-assisted rehabilitation, exoskeletons, and polishing. Compared to related results, the contribution of this paper lies in the design of the adaptive impedance learning controller for SEA-driven compliant robots to obtain variable impedance regulations without interactive force sensing.

## 2. Robot Dynamics

The considered compliant robot has the following dynamics:(1)$$\begin{array}{cc} \; & M\left(q\right)\ddot{q}+C\left(q,\dot{q}\right)\dot{q}+G\left(q\right)=K\left(\theta -q\right)+\tau_{en},\\ \; & B\ddot{\theta }+K\left(\theta -q\right)=\tau \end{array}$$
where $q\in R^{n}$ and $\theta \in R^{n}$ denote the positions of the rigid-link robot and the SEA, respectively; M(q) and *B* denote the inertial matrices; $C(q,\dot{q})$ denotes the Coriolis and centrifugal matrix; G(q) is the gravity torque; *K* is the stiffness matrix for the SEA; $\tau_{en}$ denotes the interaction force between the robot and its environment; and τ is the system control input.

**Property** **1.**
*M(q) and B are symmetric and positive definite matrices that satisfy*

(2)
$$\begin{array}{c} \sigma_{1}I\leq M\left(q\right)\leq \sigma_{2}I \end{array}$$

*where $\sigma_{1}$ and $\sigma_{2}$ are positive constants.*


**Property** **2.**
*$\dot{M}\left(q\right)-2C\left(q,\dot{q}\right)$ is a skew symmetric, i.e.,*

(3)
$$\begin{array}{c} \xi^{T}\left(\dot{M}\left(q\right)-2C\left(q,\dot{q}\right)\right)\xi =0,\;\forall \xi \in R^{n}. \end{array}$$



**Property** **3.**
*The robot dynamics have the following parameterized form*

(4)
$$\begin{array}{c} M\left(q\right)\phi_{1}+C\left(q,\dot{q}\right)\phi_{2}+G\left(q\right)=Y\left(\phi_{1},\phi_{2},q,\dot{q}\right)W \end{array}$$

*where W is a constant vector and contains unknown parameters.*


**Remark** **1.**
*The model in (1) derived by Spong [18] takes a balance between the complexity and physical validity by neglecting the inertial coupling between the link-side dynamics and the motor. The viability of the model in (1) has been demonstrated for compliant robots with SEAs [19].*


Denote $q_{d}$ as the desired trajectory of the robot in the interaction. Define the tracking error *e* as
(5)$$\begin{array}{c} e_{1}=q_{d}-q. \end{array}$$
As proven and presented in [17], the robot–environment interaction force can be expanded as
(6)$$\begin{array}{cc} \tau_{en}= & K_{s}e_{1}+K_{d}{\dot{e}}_{1} \end{array}$$
where $K_{s}=\mathrm{diag}\left\{K_{si}\right\}$, and $K_{d}=\mathrm{diag}\left\{K_{di}\right\}$ denote the stiffness and damping terms in the interaction, respectively. Denote $Q_{e}=\left[-\mathrm{diag}\left\{e\right\},-\mathrm{diag}\left\{\dot{e}\right\}\right]$ and $V={\left[K_{s1},\cdots ,K_{sn},K_{d1},\cdots ,K_{dn}\right]}^{T}$. Then, the force $\tau_{en}$ can be expressed as
(7)$$\begin{array}{cc} -\tau_{en}= & Q_{e}V. \end{array}$$

The objective of this paper is to design model-based adaptive impedance learning control using differential adaptation to estimate the impedance profiles in $Q_{e}$ so that the tracking error $e_{1}$ and impedance estimation errors are uniformly ultimately bounded (UUB) without the measurement of the interactive force $\tau_{en}$.

## 3. Impedance Learning-Based Interaction Control

This section presents an impedance learning-based adaptive interaction control strategy for the considered compliant robot using the CFAB approach. The control design procedure is stated as follows:

*Step 1:* Define the error $e_{2}$ as
(8)$$\begin{array}{c} e_{2}={\dot{e}}_{1}+k_{1}e_{1} \end{array}$$
where $k_{1}$ is a positive parameter. Based on (1), the dynamics of $e_{2}$ can be stated as
(9)$$\begin{array}{cc} M\left(q\right){\dot{e}}_{2}= & -C\left(q,\dot{q}\right)e_{2}+M\left(q\right)\left({\ddot{q}}_{d}+k_{1}{\dot{e}}_{1}\right)+C\left(q,\dot{q}\right)\left(\dot{q}+e_{2}\right)\\ \; & +G\left(q\right)-K\left(\theta -q\right)-\tau_{en}\\ = & -C\left(q,\dot{q}\right)e_{2}+Y_{e}W+Q_{e}V-K\left(\theta -q\right)-\tau \end{array}$$
where $Y_{e}≜Y\left({\ddot{q}}_{d}+k_{1}{\dot{e}}_{1},\dot{q}+e_{2},q,\dot{q}\right)$.

Design the virtual control $\alpha_{1}$ as
(10)$$\begin{array}{c} \alpha_{1}=K^{-1}\left(Kq+k_{2}e_{2}+Y_{e}\hat{W}+Q_{e}\hat{V}\right) \end{array}$$
where $k_{2}$ is a positive control gain and $\hat{W}$ and $\hat{V}$ are the estimators of *W* and *V*, respectively. The estimators are updated by
(11)$$\begin{array}{cc} \; & \dot{\hat{W}}=\gamma_{1}Y_{e}^{T}e_{2},\\ \; & \dot{\hat{V}}=\gamma_{2}Q_{e}^{T}e_{2} \end{array}$$
where $\gamma_{1}$ and $\gamma_{2}$ are the positive learning rates.

Pass $\alpha_{1}$ through the following command filter
(12)$$\left[\begin{array}{c} {\dot{\delta }}_{1}\\ {\dot{\delta }}_{2} \end{array}\right]=\left[\begin{array}{cc} 0 & I\\ -\omega^{2}I & -2\xi \omega I \end{array}\right]\left[\begin{array}{c} \delta_{1}\\ \delta_{2} \end{array}\right]+\left[\begin{array}{c} 0\\ \omega^{2}\alpha_{1} \end{array}\right]$$
where ω and ξ∈R are the frequency and the damping ratio, respectively.

Define $\alpha_{1}^{c}=\delta_{1}$, ${\dot{\alpha }}_{1}^{c}=\delta_{2}$, and
(13)$$\begin{array}{c} {\tilde{\alpha }}_{1}=\alpha_{1}^{c}-\alpha_{1}. \end{array}$$

Substituting (10) and (13) into (9) yields
(14)$$\begin{array}{cc} M\left(q\right){\dot{e}}_{2}= & -C\left(q,\dot{q}\right)e_{2}-k_{2}e_{2}+K{\tilde{\alpha }}_{1}+Y_{e}\tilde{W}+Q_{e}\tilde{V} \end{array}$$
where $\tilde{W}=W-\hat{W}$ and $\tilde{V}=V-\hat{V}$.

*Step 2:* For the SEA, define the errors $e_{3}$ and $e_{4}$ as
(15)$$\begin{array}{cc} \; & e_{3}=\alpha_{1}^{c}-\theta ,\\ \; & e_{4}={\dot{e}}_{3}+k_{3}e_{3}. \end{array}$$
From (1), the dynamics of $e_{4}$ can be presented as
(16)$$\begin{array}{c} B{\dot{e}}_{4}=K\left(\theta -q\right)-\tau +B\left({\dot{\delta }}_{2}+k_{3}{\dot{e}}_{3}\right). \end{array}$$
Design the control input τ as
(17)$$\begin{array}{c} \tau =K\left(\theta -q\right)+k_{4}e_{4}-B\left({\dot{\delta }}_{2}+k_{3}{\dot{e}}_{3}\right) \end{array}$$
where $k_{4}>0$. Then,
(18)$$\begin{array}{c} B{\dot{e}}_{4}=-k_{4}e_{4}. \end{array}$$

**Remark** **2.**
*The use of the command filter in (12) can decrease computational complexity and can avoid the requirement of the high derivatives of positions in conventional backstepping control of SEA-driven robots.*


**Lemma** **1**([20])**.**
*Consider the command filter in (12) on t∈[0,T), with T being a finite value. Given a small $\epsilon \in R^{+}$, there exists a sufficiently large ω such that $\left|\right|{\tilde{\alpha }}_{1}\left|\right|\leq \epsilon$ on t∈[0,T).*

**Theorem** **1.**
*Design the impedance learning-based adaptive interaction controller in (17) with the learning law in (11) for the considered compliant robot dynamics in (1). The tracking error $e_{1}$ and the estimation errors $\tilde{V}$ and $\tilde{\theta }$ are semiglobally uniformly ultimately bounded (SUUB).*


**Proof.** Consider the following Lyapunov function candidate
(19)$$\begin{array}{c} L=\frac{1}{2}e_{2}^{T}M\left(q\right)e_{2}+\frac{1}{2}e_{4}^{T}Be_{4}+\frac{1}{2\gamma_{1}}{\tilde{W}}^{T}\tilde{W}+\frac{1}{2\gamma_{2}}{\tilde{V}}^{T}\tilde{V}. \end{array}$$Taking the time derivative of *L* and substituting (14) and (16), one can obtain
(20)$$\begin{array}{cc} \dot{L} & =-k_{2}e_{2}^{T}e_{2}-\frac{1}{2}e_{2}^{T}\left(\dot{M}\left(q\right)-2C\left(q,\dot{q}\right)\right)e_{2}+e_{2}^{T}K{\tilde{\alpha }}_{1}\\ \; & +e_{2}^{T}Y_{e}\tilde{W}+e_{2}^{T}Q_{e}\tilde{V}-k_{4}e_{4}^{T}e_{4}-{\tilde{W}}^{T}\gamma_{1}\dot{\hat{W}}-{\tilde{V}}^{T}\gamma_{2}\dot{\hat{V}} \end{array}$$
From Property 2 and the update laws in (11),
(21)$$\begin{array}{c} \dot{L}=-k_{2}e_{2}^{T}e_{2}-k_{4}e_{4}^{T}e_{4}+e_{2}^{T}K{\tilde{\alpha }}_{1}. \end{array}$$According to Lemma 1, if the parameter ω chosen is sufficiently large, $\left|\right|{\tilde{\alpha }}_{1}\left|\right|\leq \epsilon$ on [0,T). Using Young’s inequality,
(22)$$\begin{array}{cc} e_{2}^{T}K{\tilde{\alpha }}_{1} & \leq \frac{k_{2}}{2}e_{2}^{T}e_{2}+\frac{1}{2k_{2}}{\tilde{\alpha }}_{1}^{T}K^{T}K{\tilde{\alpha }}_{1}\\ \; & \leq \frac{k_{2}}{2}e_{2}^{T}e_{2}+\frac{\epsilon^{2}k_{d}}{2k_{2}} \end{array}$$
where $k_{d}=\lambda_{max}\left(K^{T}K\right)$.Based on (21) and (22), one can obtain
(23)$$\begin{array}{c} \dot{L}\leq -\frac{k_{2}}{2}e_{2}^{T}e_{2}-k_{4}e_{4}^{T}e_{4}+\frac{\epsilon^{2}k_{d}}{2k_{2}},\;\forall t\in \left[0,T\right) \end{array}$$
which implies
(24)$$\begin{array}{c} L\left(t\right)\leq L\left(0\right)+\frac{\epsilon^{2}k_{d}}{2k_{2}},\;\forall t\in \left[0,T\right). \end{array}$$Based on Lemma 1 and (24), we can conclude that $\left|\right|{\tilde{\alpha }}_{1}\left|\right|\leq \epsilon$ can be satisfied for t∈[0,∞) if the parameter ω chosen is sufficiently large. Given the initial values for the closed-loop control system, the inequality in (23) is satisfied for t∈[0,∞) if the control parameters are properly chosen. Therefore, the proposed impedance learning-based adaptive controller makes the closed-loop control system SUUB.    □

## 4. Simulation Results

Simulations are implemented on the compliant robot arm in (1) with
(25)$$\begin{array}{cc} \; & Y_{e}=\left[{\ddot{q}}_{d}+k_{1}{\dot{e}}_{1},\sin\left(q\right),\dot{q}\right],\;Q_{e}=\left[-e_{1},-{\dot{e}}_{1}\right], \end{array}$$
(26)$$\begin{array}{cc} \; & W={\left[1,-4.9,1\right]}^{T},\;V=\left[-10,-3\right],\;K=20. \end{array}$$
For the considered robot, the initial value is chosen as $q\left(0\right)=\dot{q}=\theta \left(0\right)=\dot{\theta }=0$, and the control parameters for the proposed impedance learning-based robot adaptive control in (17) are chosen as $k_{1}=2$, $k_{2}=5$, $k_{3}=3$, $k_{4}=6$, ω=15, ξ=0.8, and $\gamma_{1}=12,\;\gamma_{2}=10$.

In the simulation, a regulation problem and a tracking problem are considered as two cases, where $q_{d}=0.7\mathrm{rad}$ for Case 1 and $q_{d}=0.2+0.3\cos\left(\pi t/6\right)$ for Case 2. The simulation results in Case 1 and Case 2 are presented in Figure 1, Figure 2 and Figure 3 and Figure 4, Figure 5 and Figure 6, respectively.

In Case 1, by using the proposed controller shown in Figure 3, the regulation error $e_{1}$ in Figure 1 is very close to zero after 10 s. In Figure 2, it can be seen that although $\tilde{W}$ and $\tilde{V}$ are not close to zero owing to not enough excitation and the coupling between the robotic parameters’ uncertainties and the impedance’s uncertainties, the robotic parameters’ estimation error $\tilde{W}$ and the impedance profiles’ estimation error $\tilde{V}$ are significantly decreased after 5 s and the force estimation errors $Y_{e}\tilde{W}$ and $Q_{e}\tilde{V}$ are very close to zero after 10 s.

In Case 2, the proposed impedance learning controller shown in Figure 6 renders the tracking error $e_{1}$ in Figure 4 ultimately close to zero. In Figure 5, it can be seen that the robotic parameter estimation error $\tilde{W}$ and the impedance profile estimation error $\tilde{V}$ are highly decreased, $Q_{e}\tilde{V}$ is close to zero, and $Y_{e}\tilde{W}$ is bounded but not close to zero. The reason is that $Q_{e}\tilde{V}$ plays a more important role in $e_{1}$ than $Y_{e}\tilde{W}$ and $\tilde{V}$ receives more excitation.

The above simulation results illustrate the effectiveness of the proposed impedance learning-based controller in (17) and the adaptive impedance learning in (11). The proposed controller can ensure that robot impedance is more close to human impedance than impedance control with constant impedance profiles.

## 5. Conclusions

The variable impedance control of robots can improve human–robot interaction performance through the regulation of the impedance of robots to adjust the motions of human limbs. However, impedance variation affects the control stability of robots. Based on impedance learning, this paper has designed an adaptive controller of SEA-driven robots for human–robot interaction using the command filter-based adaptive backstepping approach. Adaptive estimators have been designed to approximate the robot modeling uncertainty and impedance parameters of interaction force. We have validated the practical control stability through theoretical analysis and showed the control effectiveness through simulations. The designed impedance learning control provides variable robot impedance regulation without interactive force sensing. By exploiting the advantages of impedance learning control and compliance actuators, this paper improves the safety and compliance of robot–environment interactions. In this paper, we only guarantee that the control system is SUUB. Guaranteeing that the impedance learning-based control is asymptotically stable and improving impedance estimation performance are our future research directions.

## Figures and Tables

**Figure 1 sensors-22-09740-f001:**
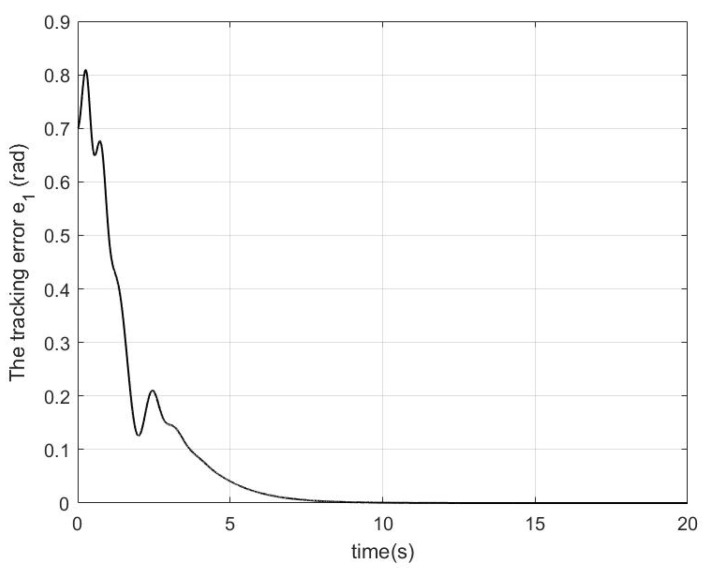
The performance of the tracking error $e_{1}$ in Case 1.

**Figure 2 sensors-22-09740-f002:**
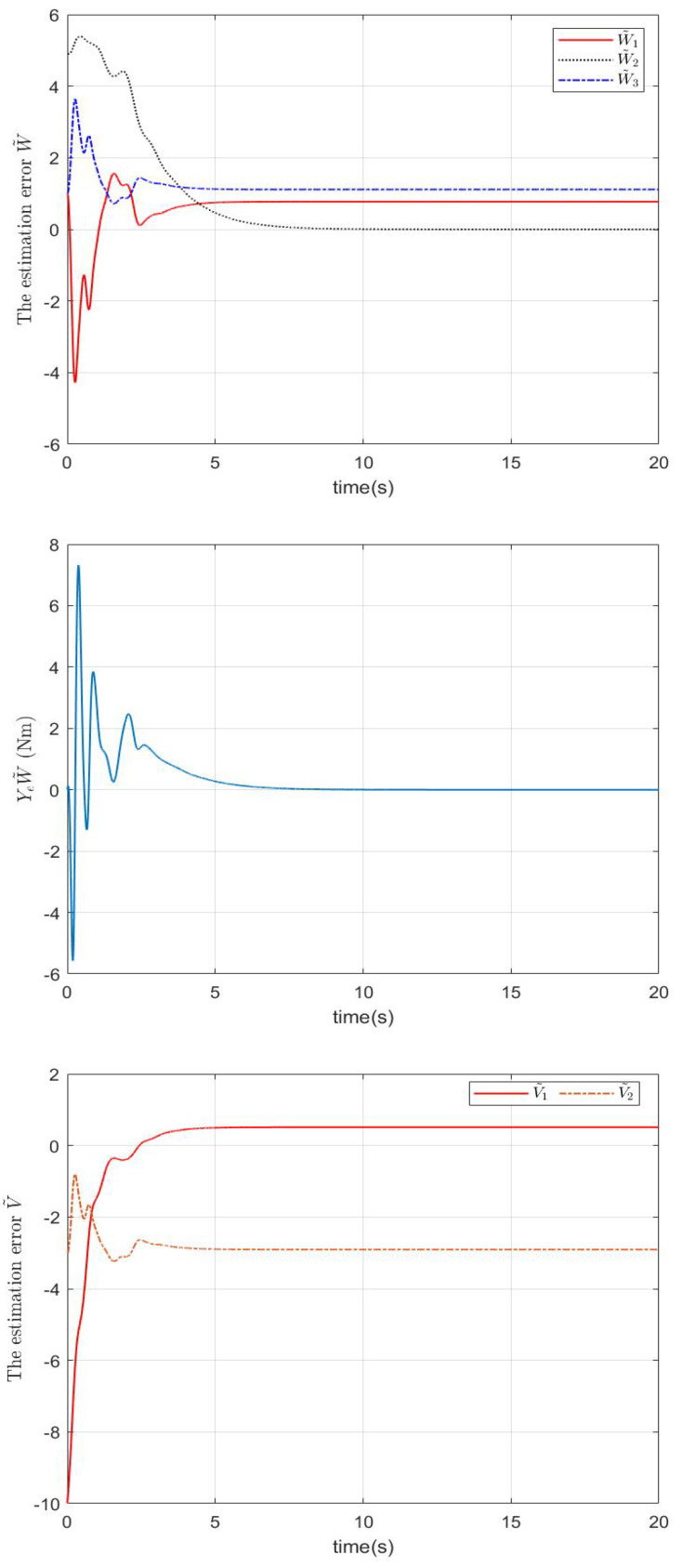
The estimation errors $\tilde{W}$, $Y_{e}\tilde{W}$, $\tilde{V}$, and $Q_{e}\tilde{V}$ in Case 1.

**Figure 3 sensors-22-09740-f003:**
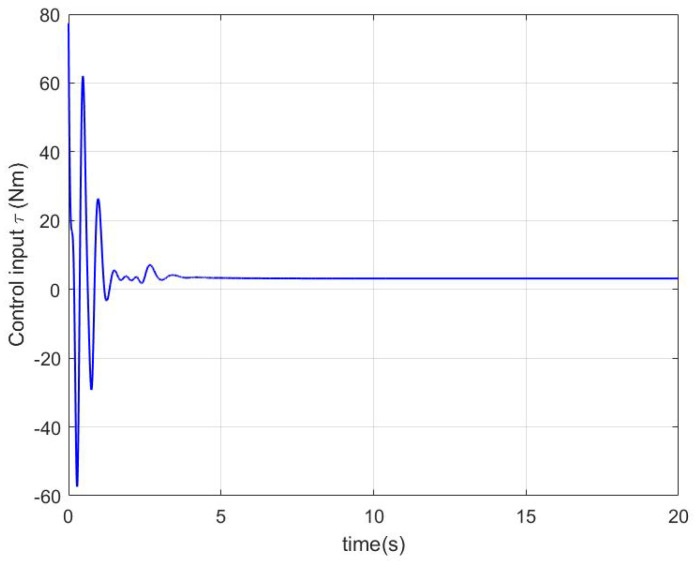
The control input of (17) in Case 1.

**Figure 4 sensors-22-09740-f004:**
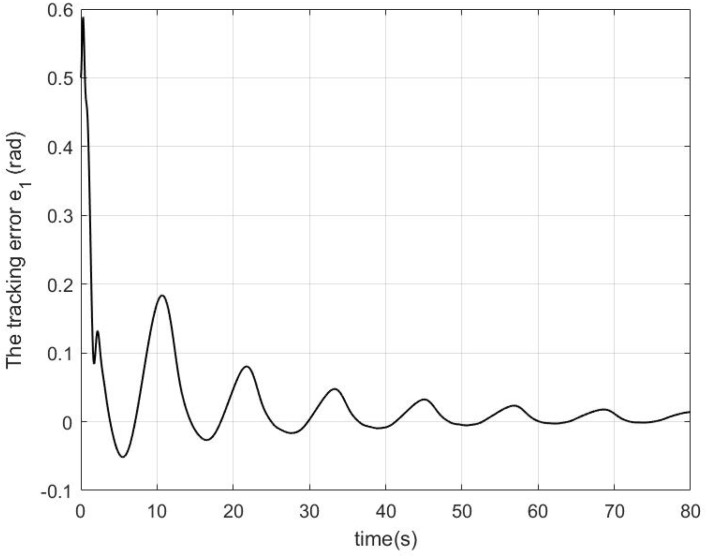
The performance of the tracking error $e_{1}$ in Case 2.

**Figure 5 sensors-22-09740-f005:**
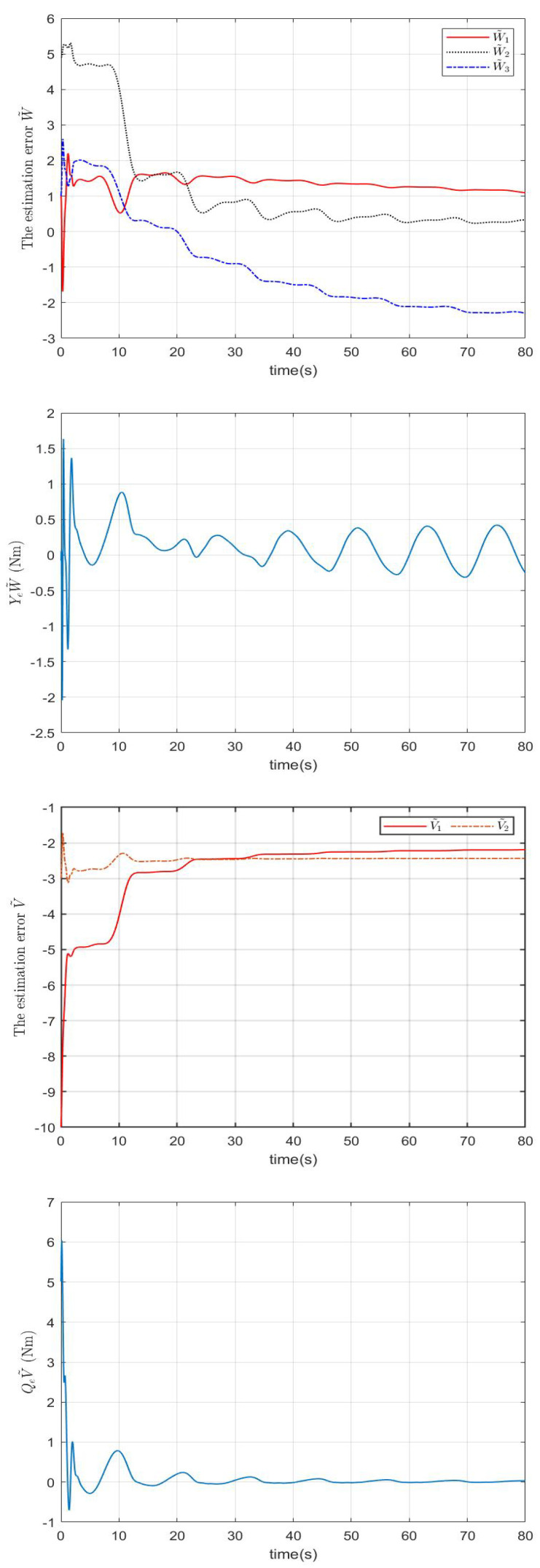
The estimation errors $\tilde{W}$, $Y_{e}\tilde{W}$, $\tilde{V}$, and $Q_{e}\tilde{V}$ in Case 2.

**Figure 6 sensors-22-09740-f006:**
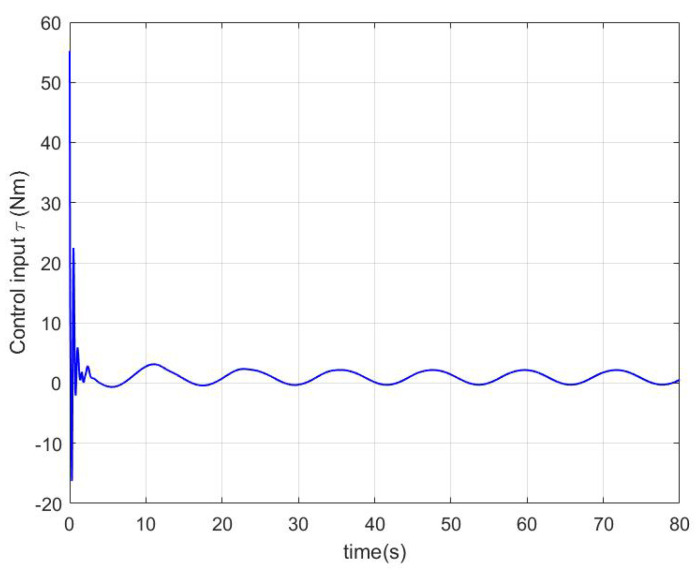
The control input of (17) in Case 2.

## Data Availability

Not applicable.

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
