# Peer review of "Adaptive Interaction Control of Compliant Robots Using Impedance Learning"

_sensors, 2022, doi:10.3390/s22249740_

Round 1

Reviewer 1 Report

The paper presents an exciting topic of impedance learning-based adaptive control strategy for series two elastic actuators (SEAs)-driven compliant robots without measurement of robot-environment three interaction force. The paper presentation is inferior and needs significant revision. 

1- The dynamic model of the considered robot is not well discussed. 

2- The results are poorly presented. For example, some figures have no y-axis title, and some have no legends. This makes the readability of the paper very bad. 

3- The scientific contribution of the paper is not clear. 

Author Response

Responses to Reviewer #1

Comment: The paper presents an exciting topic of impedance learning-based adaptive control strategy for series two elastic actuators (SEAs)-driven compliant robots without measurement of robot-environment three interaction force. The paper presentation is inferior and needs significant revision. 

Response: The authors would like to express their sincere appreciation to the reviewer for his/her constructive comments and suggestions, and his/her time and efforts spent in helping us to improve the quality and presentation of the paper. We have highly improved the quality of this paper according to your valuable comments.

Comment 1: The dynamic model of the considered robot is not well discussed. 

Response: Thank you for the helpful comment. We have added Remark 1 to discuss the dynamic model in (1).

“Remark 1: The model (1) derived by Spong [18] takes a balance between complexity and physical validity by neglecting the inertial coupling between the link-side dynamics and the motor. The viablity of the model (1) has been demonstrated for compliant robots 66 with SEA [19].”

Comment 2: The results are poorly presented. For example, some figures have no y-axis title, and some have no legends. This makes the readability of the paper very bad. 

Response: Thank you for the helpful comment. We have improved the presentation of the results by adding more simulation analysis and adding missed y-axis title and legends.

Comment 3: The scientific contribution of the paper is not clear. 

Response: Thank you for the constructive comment. In Introduction, we have clearly stated the paper’s contribution with blue-background.

“Recently, model-based impedance learning control strategies [15–17] were developed for robot-environment interaction and validated in repetitive tasks with stability guarantee. The control approach can provide variable impedance regulation for robots without the requirement of interaction force sensing. However, the existing model-based impedance learning controllers mainly focus on rigid-link robots. The extension of model-based impedance learning control to SEA-driven compliant robots is not directly, since the introduction of SEA highly increases control design complexity and makes the control system from a second-order fully actuated system to a fourth-order underactuated system.

“Based on the above analysis, designing model-based impedance learning control for SEA-driven robots can exploit the advantages of passive compliant devices and active compliance control to improve robot-environment interaction performances, but no result on this research topic was developed by now.”

“The proposed control strategy can be applied in categories of robot-environment interaction including robot-assisted rehabilitation exoskeletons, and polishing. Compared with the related results, the contribution of this paper lies in the design of the adaptive impedance learning controller for SEA-driven compliant robots to obtain variable impedance regulation without interactive force sensing.”

Reviewer 2 Report

This paper proposes an impedance learning-based adaptive control strategy for SEA-driven compliant robots without measurement of interactive force. The proposed controller can provide variable impedance regulation for the considered compliant robot. Theoretical analysis and simulations were presented to validate the control effectiveness. The work is solid and interesting to read. But some revisions are required to further improve the quality of this paper.

1)      Potential use and application ranges of the proposed method should be highlighted in order to arise readers’ interest.

2)      In the dynamics of compliant robots, there exists possible inertial coupling between the robot and the compliant actuator. Some analysis should be given to show the generality of the dynamics in (1).

3)      In simulations, the figures should be explained with more details right after their presentation.

4)      There are some typos and grammatical errors. Please correct them.

Author Response

Responses to Reviewer #2

Comment: This paper proposes an impedance learning-based adaptive control strategy for SEA-driven compliant robots without measurement of interactive force. The proposed controller can provide variable impedance regulation for the considered compliant robot. Theoretical analysis and simulations were presented to validate the control effectiveness. The work is solid and interesting to read. But some revisions are required to further improve the quality of this paper.

Response: The authors would like to express their sincere appreciation to the reviewer for his/her constructive comments and suggestions, and his/her time and efforts spent in helping us to improve the quality and presentation of the paper. Thank you for the positive comment.

Comment 1: Potential use and application ranges of the proposed method should be highlighted in order to arise readers’ interest.

Response: Thank you for the helpful comment. We have added the potential use and application ranges in Introduction.

“The proposed control strategy can be applied in categories of robot-environment interaction including robot-assisted rehabilitation, exoskeletons, and polishing.”

Comment 2:  In the dynamics of compliant robots, there exists possible inertial coupling between the robot and the compliant actuator. Some analysis should be given to show the generality of the dynamics in (1).

Response: Thank you for the helpful comment. We have added Remark 1 to discuss the dynamic model in (1).

“Remark 1: The model (1) derived by Spong [18] takes a balance between complexity and physical validity by neglecting the inertial coupling between the link-side dynamics and the motor. The viablity of the model (1) has been demonstrated for compliant robots 66 with SEA [19].”

Comment 3:    In simulations, the figures should be explained with more details right after their presentation.

Response: Thank you for the valuable comment. We have improved the presentation of the simulation results.

Comment 4: There are some typos and grammatical errors. Please correct them.

Response: Thank you for the helpful comment. We have improved the presentation by correcting typos and grammatical errors.

Reviewer 3 Report

This interesting paper presents an impedance learning-based adaptive control strategy for series elastic actuators (SEAs)-driven compliant robots without measurement of robot-environment

interaction force.

Lack of problem statements, the purpose of this paper is equivocal. No significant contributions for knowledge and science as long as just a simulation. Authors are fail to show novelty and research contribution.

What kind of software do you use for simulation?

+ Mathematic modelling

- Only simulation

Please consider these articles for your references

Sasaki, M., Honda, N., Njeri, W., Matsushita, K., & Ngetha, H. (2020). Gain tuning using neural network for contact force control of flexible arm. JOURNAL OF SUSTAINABLE RESEARCH IN ENGINEERING, 5(3), 138-148.

Njeri, W., Sasaki, M., & Matsushita, K. (2019). Gain tuning for high-speed vibration control of a multilink flexible manipulator using artificial neural network. Journal of Vibration and Acoustics, 141(4).

Author Response

Responses to Reviewer #3

Comment: This interesting paper presents an impedance learning-based adaptive control strategy for series elastic actuators (SEAs)-driven compliant robots without measurement of robot-environment interaction force.

 Responses: The authors would like to express their sincere appreciation to the reviewer for his/her constructive comments and suggestions, and his/her time and efforts spent in helping us to improve the quality and presentation of the paper. Thank you for the positive comment.

Comment 1: Lack of problem statements, the purpose of this paper is equivocal. No significant contributions for knowledge and science as long as just a simulation. Authors are fail to show novelty and research contribution.

Response: Thank you for the helpful comment.

(i)In the last paragraph of Section 2, we have added the objective of this paper.

“The objective of this paper is to design model-based adaptive impedance learning control using differential adaptation to estimate the impedance profiles in Qe, such that the tracking error e1 and impedance estimation errors are uniformly ultimatly bounded (UUB) without the measurement of the interactive force τen.”

(ii)In Introduction, we have clearly stated the contribution and innovation of this paper with blue-background.

“Recently, model-based impedance learning control strategies [15–17] were developed for robot-environment interaction and validated in repetitive tasks with stability guarantee. The control approach can provide variable impedance regulation for robots without the requirement of interaction force sensing. However, the existing model-based impedance learning controllers mainly focus on rigid-link robots. The extension of model-based impedance learning control to SEA-driven compliant robots is not directly, since the introduction of SEA highly increases control design complexity and makes the control system from a second-order fully actuated system to a fourth-order underactuated system.

“Based on the above analysis, designing model-based impedance learning control for SEA-driven robots can exploit the advantages of passive compliant devices and active compliance control to improve robot-environment interaction performances, but no result on this research topic was developed by now.”

“The proposed control strategy can be applied in categories of robot-environment interaction including robot-assisted rehabilitation exoskeletons, and polishing. Compared with the related results, the contribution of this paper lies in the design of the adaptive impedance learning controller for SEA-driven compliant robots to obtain variable impedance regulation without interactive force sensing.”

Comment 2: Please consider these articles for your references

Sasaki, M., Honda, N., Njeri, W., Matsushita, K., & Ngetha, H. (2020). Gain tuning using neural network for contact force control of flexible arm. JOURNAL OF SUSTAINABLE RESEARCH IN ENGINEERING, 5(3), 138-148.

Njeri, W., Sasaki, M., & Matsushita, K. (2019). Gain tuning for high-speed vibration control of a multilink flexible manipulator using artificial neural network. Journal of Vibration and Acoustics, 141(4).

Response: Thank you for the helpful comment. The mentioned references make significant contributions in force control and vibration control of flexible arm. We have cited these results in references [10] and [11].

Round 2

Reviewer 1 Report

The authors address all comments. In my opinion, the paper lacks the experimental part to be accepted in the Journal. If the authors can add an experiment to validate the interesting theoretical results, it will be more presentable. 

Thank you

Author Response

Comment: The authors address all comments. In my opinion, the paper lacks the experimental part to be accepted in the Journal. If the authors can add an experiment to validate the interesting theoretical results, it will be more presentable. 

Response: The authors would like to express their sincere appreciation to the reviewer for his/her constructive comments and suggestions, and his/her time and efforts spent in helping us to improve the quality and presentation of the paper.

We agree with you that the presentation of this paper will be highly improved if an experiment is added. However, we don’t have the experiment condition at present. We would like to do the related experiment in the future. Thank you very much.

Reviewer 3 Report

The authors have revised the paper properly. However, a more in-depth and comprehensive discussion will contribute more to knowledge. Justification of what has been found in this research and directions for future research needs to be conveyed.

Author Response

Comment: The authors have revised the paper properly. However, a more in-depth and comprehensive discussion will contribute more to knowledge. Justification of what has been found in this research and directions for future research needs to be conveyed.

Response: The authors would like to express their sincere appreciation to the reviewer for his/her constructive comments and suggestions, and his/her time and efforts spent in helping us to improve the quality and presentation of the paper.

At the last paragraph of Conclusions, we have added the finding in this research and future research.

‘The designed impedance learning control provides variable robot impedance regulation without interactive force sensing. By exploiting the advantages of impedance learning control and compliance actuators, this paper improves the safety and compliance for robot-environment interaction. However, in this paper, we only guarantee the control system be SUUB. How to guarantee impedance learning-based control to be asymptotically stable and how to improve impedance estimation performance is in our future research directions.’